# Motivations to Collect: How Consumers Are Socialized to Build Product Collections

Jennifer Johnson Jorgensen *, Katelyn Sorensen and Melisa Spilinek

Department of Textiles, Merchandising & Fashion Design, University of Nebraska-Lincoln, Lincoln, NE 68583-0802, USA
* Correspondence: jbjorgensen@unl.edu

**Abstract:** Most people have collected products at some point in their lives; however, little is known about how people are socialized to collect. This mixed methods study recruited 213 participants to explain and explore the influences of family, friends, romantic partners, and online social media on the continued intention to build product collections. Qualitative findings revealed a clear pattern of familial influences when participants shared how their collections started. When starting collections, participants acquired products through either personal interest in the products or receiving gifts from family members. However, quantitative results indicate that friends, romantic partners, and social media have a greater influence after the product collection has started. The results and findings of this study also guide an adaptation of the consumer socialization theory.

**Keywords:** product collection; consumer socialization; family; social media; friends





## 1. Introduction

Product collections are nothing new, as most people have collected either physical or digital products at some point in their lives (Harrison et al. 2017). Currently, it is believed that approximately 33% of the U.S. population collects (Mueller 2019). People collect a wide range of things, as most products can be removed from everyday use to build a collection (Carey 2008), but the people who choose to collect may be motivated by different influences. Some collectors may not have intended to start their collections but acquired a product and added more over time. Other collectors may have decided to create collections and purchase such products (Carey 2008; Gao et al. 2014). Despite how collections were started, new consumption habits driven by online social media and retailing may have changed how people have been socialized to collect products.

According to Belk (1995), collecting is "the process of actively, selectively, and passionately acquiring and possessing things removed from ordinary use and perceived as part of a set of non-identical objects or experiences" (Belk 1995; Carey 2008). Many collectors do not intend to collect and start the process from the "bottom–up", while others have a top–down approach and intend to collect one day (Gao et al. 2014). Collectors are driven by the goal of accumulating many items to create a set (Zonneveld and Biggemann 2014) and gather objects to become part of that set for many reasons, the most common being for satisfaction or financial gain (Carey 2008; Lafferty et al. 2013; Mueller 2019). Turunen and Leipämaa-Leskinen (2015) found collectors have a solid emotional connection with their possessions. Beyond a specific type of product, brands may also be collected that invoke nostalgia that aligns with the specific brand's values (Kessous et al. 2015; Zonneveld and Biggemann 2014). When people feel nostalgia toward a brand, it has been found that they are more likely to collect than those that do not spur nostalgic feelings (Kessous et al. 2015).

Family members and peers (or friends) have been found to influence people to start collecting (Lafferty et al. 2013; Nordsletten and Mataix-Cols 2012). In particular, family members were found to impact the type of product collected, typically driven by the receiver's interests (Gao et al. 2014). The gifting of products without the intention to add to

one's collection has also been determined as a method by which collections grow over time. Conversely, collections may be inherited from other family members (Gao et al. 2014).

Online technologies have led to new consumption habits, as collectors can have physical or digital product sets. For example, baseball cards can now be collected on mobile apps where collectors can buy and trade digitally (Parker 2018). Thus, how older collectors purchase products for their collections may differ from those of younger generations. More senior collectors may still prefer bricks-and-mortar stores even though the marketplace is changing to online sales (Halperin 2018; Parker 2018). Online communities and websites can also provide collectors with virtual connections to share their love of a brand or product (Kessous et al. 2015) and purchase products for their collections. These virtual connections can help solidify a commitment to building the collection.

Due to possible shifts in consumer behavior for collectors because of online technologies, the purpose of this mixed methods study was to explain the influence of family, friends, romantic partners, and online social media on the intention to continue product collections, as well as to explore how collectors are socialized to have an interest in collecting products. The consumer socialization theory guided this research, which outlines that consumer behavior is learned through observation of family, peers, and media (Moschis and Churchill 1978), as it has been found that intimate social interactions impact our brain and behavior (Mueller 2019). Thus, there is a lack of research in the current literature investigating collecting products facilitated through online technologies.

## 2. Review of the Literature

Research on collector socialization is limited, as many studies focus on either the difference between collectors and hoarders or the collection of a specific product category (Nordsletten and Mataix-Cols 2012). This study focuses on collectors, since hoarders exhibit different behaviors and motivations to acquire new items than collectors. Traditional collections, art, and brands are frequently researched subtopics of collecting.

### 2.1. Traditional Collections

Traditional collections include art, antiques, books, baseball cards, stamps, and coins (Carey 2008; Gao et al. 2014; Lafferty et al. 2013). Souvenirs from travel destinations, such as magnets, shot glasses, and postcards, are also commonly collected (Carey 2008). Beanie Babies, Cabbage Patch dolls, and Pokémon trading cards were popular in the 1990s when companies encouraged individuals to collect the entire product line. In addition, items worn by celebrities have always been popular to collect, and individuals pay high prices for these one-of-a-kind items (McIntosh and Schmeichel 2004).

Brands are collected as individuals feel nostalgia and live the brand's values (Kessous et al. 2015; Zonneveld and Biggemann 2014). Some individuals are part of an online community or create websites to share their love of the brand (Kessous et al. 2015). Kessous et al. (2015) discovered individuals who feel a brand is nostalgic are more likely to collect when compared to non-nostalgic brands. Coca-Cola is the most collected brand globally, and collectors feel nostalgia, patriotism, and pride towards the brand (Kessous et al. 2015). Zonneveld and Biggemann (2014) researched the Crown Lynn brand and the motivations behind collecting brand products. The main reasons were the brand's country of origin and heritage (Zonneveld and Biggemann 2014). Disney is another famous international brand that reminds individuals of their childhood. For example, Disney releases a limited number of collectible pins, and they have created a vast range of collectors (Lafferty et al. 2013).

Apparel is commonly collected, representing a piece of the past and bringing one back to life and the culture of the time (Zonneveld and Biggemann 2014). Many clothing collectors stated they wanted their collection to be used for education and for all the pieces to stay together. Traditional clothing collections can also include vintage garments and accessories. Clothing collectors sometimes wear pieces of their collections or aim to preserve each item. Historians want pieces representative of the time frame, and such

collectors strive for excellence or museum-quality pieces (de la Haye 2018). Luxury brands, fashion houses, and fashion designers have recently started to keep one of every item made to create a collection that displays the company's history. This type of collecting is known as commercial collecting.

Collectors deem it essential to share their knowledge with others and understand the pieces in their collections (AXA Art 2014). To gain knowledge, collectors visit trade shows, events, galleries, online sites, and trade journals, and have contacts with the same interests (AXA Art 2014). Thus, opportunities to connect with fellow collectors in an online space may have an increased impact on purchasing products for a collection.

### 2.2. Digital Collections

Digital collecting has become popular in recent years, including logging steps, food intake, digital diaries, and social media sites. Wearable technology allows users to track their health every second of the day. Another form is obtaining electronic products or pictures and creating a digital collection. A social media site, Pinterest, allows individuals to save items they enjoy and develop categories of collections (Harrison et al. 2017). Few companies offer digital items to interest a younger generation of collectors. Younger generations have different consumption habits, as they have been found to prefer digital collections. Panini is a company that creates collectible cards for the NBA, NFL, and FIFA soccer for people to collect in an app (Parker 2018). Virtual collections, especially non-fungible tokens (NFTs), have also increased in popularity (BlueWeave Consulting 2022).

### 2.3. Finding Products to Collect

Collectors often turn to secondary markets to find missing pieces in their collections, increasing the product's social value. When numerous people collect a specific product type, it helps create a set of standards and a social network. A community of collectors that enjoy the same things creates social acceptance, and conferences and trade shows are often put together to help each other complete sets (Carey 2008). Secondary markets include online auction sites, bricks-and-mortar stores, television shows, and trade shows (Gao et al. 2014). Secondary markets such as television shows and eBay have also changed buying patterns for collectors (Benson 2017).

eBay is an online auction site that has changed how people collect, as they can quickly obtain items they need. A downside of online auction sites is that they can create an over-supply of collectibles that decreases the rarity of collecting (Carey 2008). The e-commerce of collectibles has led to the emergence of collector organizations and online communities (Gao et al. 2014). Many channels on television sell collectibles and have encouraged collecting (McIntosh and Schmeichel 2004). For example, one successful show is the BBC's Antique Roadshow, which has aired for 33 seasons with over 6 million viewers (Nordsletten and Mataix-Cols 2012).

### 2.4. Motivations to Collect

According to Belk (1995), "collecting as a materialistic luxury consumption that yields utility to the individual consumer, presents a selfish opportunity cost for others in the household, and represents marker goods that announce social class" (Belk 1995; Carey 2008). Collecting symbolizes the owner's culture, self-identity, values, and relationship to the material world (Zonneveld and Biggemann 2014). McIntosh and Schmeichel (2004) note four types of collectors: (1) passionate collectors who are irrational, (2) inquisitive collectors who collect for a profit, (3) hobbyists who obtain items for enjoyment, and (4) expressive collectors who collect for their self-reflection. Zonneveld and Biggemann (2014) identify similar motivations to collect, such as gaining pleasure and happiness when a new item is acquired, reminding them of their past and current life experiences, or displaying their social status. Collecting aims to complete a set that is not too difficult or easy to acquire, discouraging individuals from giving up (Carey 2008). Nostalgia is another

popular motivation defined as an era, place, or person from an idealized version of the past (Zonneveld and Biggemann 2014). Sometimes, people think of the past as a simpler time, and childhood memories are why collectors collect items from a particular era (de la Haye 2018; Zonneveld and Biggemann 2014).

### 2.5. How Individuals Are Socialized to Learn How to Collect

For most individuals, collecting starts at a young age and is influenced by one's parents and peers (Lafferty et al. 2013; Nordsletten and Mataix-Cols 2012). Parents influence collecting by purchasing a product from a category their child can add. Collecting can also be started by acquiring an item as a gift every year and is not intended to start a set. Collecting is also influenced by many factors, such as inherited objects (Gao et al. 2014). An equal number of men and women are collectors, and more men continue collecting throughout their lives compared to women. Some people maintain their collections over their lifetime, and others pick up the interest during retirement. In retirement, collectors can dedicate more time to collecting, adding to their set, and being experts on their collection. Collectors gain knowledge and become experts about their collections by researching their products' origin, value, condition, and insight from other experts (McIntosh and Schmeichel 2004). Many passionate collectors are willing to pay any price for an item they want and often go to events or conventions to trade objects to add to their set (Lafferty et al. 2013).

Many collectors do not think they will start a collection. It happens after acquiring a few similar items (Gao et al. 2014). Collections begin with acquiring items an individual is interested in and eventually become a collection after adding multiple pieces (de la Haye 2018). According to a study by Gao et al. (2014), when participants acquire two items, they feel justified in starting a collection. Two items are the tipping point to creating a collection since individuals must decide to start a set or find another use for the items (Gao et al. 2014; Nordsletten and Mataix-Cols 2012). The motivations to collect are to feel in control, self-identity, and relive the past (Gao et al. 2014). The collector has a psychological value in gathering new items because of the time and effort spent accumulating them (Lafferty et al. 2013; Zonneveld and Biggemann 2014). Collectors value the objects in their set and do not intend to dispose of them in the future (McIntosh and Schmeichel 2004; Nordsletten and Mataix-Cols 2012).

### 2.6. Consumer Socialization Theory

Identified as the consumer socialization theory (CST) by Moschis and Churchill (1978), "the process by which young people develop consumer-related skills, knowledge, and attitudes" (p. 599) guides the influences of external factors on purchase intentions. This study focuses on socialization agents, which can be anyone or any group with frequency, control, or primacy over the learner (Moschis and Churchill 1978, p. 600). Socialization agents primarily influence an individual's behavior and the outcomes within the CST (Bush et al. 1999). The CST was chosen for this study to understand how friends, family, romantic partners, and social media influence purchase intention through the consumers' socialization process.

### 2.7. Literature on Social Media

Many generations widely use social media. Typically, younger generations use social media to connect with friends, be entertained, and fill spare time. However, older generations have been found to have similar motivations to use social media, as Barker (2012) found that millennials and baby boomers are similar in how they use social media. Both generations have been found to interact with their peers on social media and feel optimistic about the peer groups to which they belong (Barker 2012). Consumers also engage in resonance behaviors (including 'liking' and 'sharing' on social media) at differing rates. When social ties are substantial, people are more willing to engage in resonance (Shang et al. 2017). Peer communication has also been found to relate to social media's usefulness (Harrigan et al. 2021).

Social media users also use the platform to gain information and create social value (Shang et al. 2017). Emotional and information support has been found to stem from knowledge learned from online forums and communities (Riaz et al. 2021). If people engage in a higher intensity of social media usage, perceived usefulness and purchase intention of social media positively result (Harrigan et al. 2021). The perception and use of social media are also related to the social influence of peers on Instagram, and normative beliefs have been found to impact purchase intention (Copeland and Zhao 2020). Overall, purchase decisions can be influenced by information posted and electronic word-of-mouth on social media (Shang et al. 2017; Johnson Jorgensen and Ha 2019).

**Hypothesis 1.** *Social media impacts the purchase intentions to continue a product collection.*

*2.8. Literature on Friends*

Purchases based on peer influences are believed to be connected to the number of people adopting the product within the peers' social networks. People are also more willing to purchase when mutual interactions and influences are found among friends within the social network, and purchases are more probable if more friends have purchased beforehand. The more friends someone has, the less probability of purchasing occurring. Most purchases influenced by peers are linked to one's local social circle (Zhu et al. 2016). Friendships build a stronger purchase intention than strangers, and they are more likely to purchase high-risk products from friends than strangers. High-quality friendships between buyers and sellers indicate a higher likelihood of selling high-priced and high-risk products. These high-quality friendships also indicate stronger purchase intentions than those with low-quality friendships. However, people will buy from reputable (sellers with high reviews) strangers over simple friends (Li et al. 2018). Peer groups can also connect via social media. As with any friendship, reciprocity on social media is crucial in defining relationships (Lehtinen et al. 2009). Socializing is needed to be healthy (Ristau 2011).

**Hypothesis 2.** *Friends impact the purchase intentions to continue a product collection.*

*2.9. Family*

Traditionally, children learn their consumption behaviors from their parents. Children observe interactions between household members, directly or indirectly impacting how the children learn to consume. It has also been found that acts of care and love influence how people shop for unique products, specifically grocery items (Venn et al. 2017). Joint families have been found to have a more positive intention toward online purchases than nuclear families (Bhat et al. 2021). In addition, family members have been found to impact purchase attitudes but not purchase intention of general products (Johnson Jorgensen and Ha 2019). Specific to collecting, a study by Gao et al. (2014) found that families influence collections based on the purchase of a product to fulfill their child's interest. The purchase of additional, similar products may also result (Gao et al. 2014).

**Hypothesis 3.** *Family impacts the purchase intentions to continue a product collection.*

*2.10. Literature on Romantic Partners*

Romantic partners can have an influence on consumer behavior in various ways. Yu (2011) found that the strength, satisfaction, and length of a romantic relationship have been found to predict shopping behavior. Participants who kept their romantic partners in mind while participating in the study found that the length of the relationship was linked to impulsive purchases and less exploratory shopping behavior. Conversely, participants who kept an acquaintance in mind found that romantic relationship length predicted less impulsive purchases and more exploratory behavior (Yu 2011). Similarly, when picturing a valued relationship that does not exist, participants were found to decrease indulgent products compared to participants with a valued relationship (Cavanaugh 2014). People

satisfied with their current relationship avoided conspicuous consumption (Liu et al. 2020). However, for participants who felt like they were deserving, indulgence behavior tends to unfold (Cavanaugh 2014). Women have been found to engage in conspicuous consumption that is believed to heighten attractiveness and indicate partner loyalty (Zhao et al. 2017). Feelings of loneliness also increase single people's conspicuous consumption (Liu et al. 2020). Based on the current literature on the impact of romantic relationships on consumption behavior, the following hypothesis was developed:

**Hypothesis 4.** *Romantic partners impact the purchase intentions to continue a product collection.*

### 2.11. Literature on Purchase Intention

The CST supports the idea that family, friends, and media impact attitude and purchase intention (Moschis and Churchill 1978). It has also been found that purchase intention is also impacted by the cost of the product and the individual's financial state (Parment 2013). Emotional and informational support has also been positively linked to purchase intention on social media, with information support being the most influential (Riaz et al. 2021). Purchase intention is higher when buying from a reputable stranger (one with high reviews) than from a non-reputable stranger. At the same time, reputable strangers have also been found to be more impactful than simple friends (Li et al. 2018).

Males tend to be more innovative and perceive stronger usefulness than females, linked to stronger purchase intentions. Middle-aged people have a higher perceived ease of purchasing, impacting the group's attitudes toward online purchasing (Law and Ng 2016). Baby boomers are also more likely to have relationships with specific retailers (Parment 2013). Younger consumers have been found to have higher online purchase intentions than older consumers. In addition, single people are more likely to shop online than married people (Bhat et al. 2021).

## 3. Materials and Methods

### 3.1. Research Design

This study used a convergent mixed methods approach to data collection, as quantitative and qualitative data were obtained through a survey with Likert-type scales and open-ended questions. A convergent design allows for the concurrent collection of different types of data on the same topic to more deeply understand the topic under investigation. Overall, a mixed methods approach was warranted for this study, as value can be derived from both quantitative and qualitative data to understand both the influence on collectors, as well as why and how collectors engage with their product collections.

The survey consisted of 28 Likert-type scale questions and 6 open-ended questions. The scale questions were adapted from previous studies focused on the CST, including eight questions representing the construct of social media usage and five questions for each of the constructs of family, friends, romantic partners, and purchase intention. The survey questions can be found in Appendix A. An example of a scale question includes "Pictures from online social networks encourage me to make purchases for my collections", "I ask my family for advice about buying products for my collection", and "I intend to buy products for my collection in the future that I've searched for online". These adapted questions from CST studies utilized these questions in quantitative studies analyzed quantitative data through a variety of approaches, including multiple regression, maximum likelihood estimation, and principal components analysis (Bush et al. 1999; Johnson Jorgensen and Kean 2018; Lueg et al. 2006; Mangleburg et al. 1997; Moschis and Moore 1979).

The six open-ended questions explored information about participants' specific collections, influences, and consumption behavior. Open-ended questions mainly focused on the who, what, when, and how questions and an example of one of these questions include "How did you get started in collecting?" The qualitative data were analyzed using a grounded theory approach and underwent three rounds of coding managed by MaxQDA software (MaxQDA 2022, VERBI Software, Berlin, Germany) to identify major themes.

A grounded theory approach was taken for the qualitative phase of this study. Grounded theory is known as the qualitative approach that has systematic guidelines for collecting and analyzing data to build a theory "grounded" in the data (Charmaz 2014). This specific approach was determined to be the most appropriate for this study, as collectors (as consumers) engage in a sequential process that would best lend itself to theoretical development. In line with the grounded theory approach, the researchers engaged in three rounds of coding and utilized the constant comparative method to ensure that the resulting theory represented the process (Charmaz 2014).

### 3.2. Data Collection and Analysis

Participants were recruited in March 2022 via Amazon's MTurk to take the survey through the Qualtrics platform. Amazon "workers" on the MTurk platform can access numerous surveys and tasks that they complete for varying amounts of compensation. Such "workers" or participants were able to access the survey via a list of opportunities on the MTurk website. Participants volunteered to take part in this study and were compensated USD 0.10.

Through Amazon's MTurk, 213 usable responses were collected. The participants' demographics comprised 38.2% males, 60.5% females, and the remainder chose not to specify. Most participants were between the ages of 25–34 (32.8%) and 35–44 (30.1%). Most participants identified as Caucasian (67.7%), followed by Asian (8.4%) and mixed ethnicity (8.4%) descent. A representative household income range was also obtained.

Both the quantitative and qualitative components of this study were concurrently completed. Cronbach's alpha for all scales was $0.847 > \alpha > 0.943$, and the Likert-type questions were analyzed via forward regression via SPSS. A forward regression was selected for this study, as the goal was to build a simple-to-interpret model that would be incrementally built using the most relevant variables in anticipation of widespread use across varying audiences.

## 4. Results and Findings

### 4.1. Results

All tested relationships were supported by a forward regression in SPSS, except for one. Friends, social media, and romantic partners were found to significantly impact the intention to continue product collection purchases ($R^2 = 0.685$, Adj $R^2 = 0.681$, mean square = 29.281, $F(3, 209) = 151.703$, $p > 0.05$; see Figure 1). Friends ($\beta = 0.283$, $p > 0.05$) were found to be the most impactful in making purchase decisions for collections, followed by romantic partners ($\beta = 0.250$, $p > 0.05$) and social media ($\beta = 0.214$, $p > 0.05$). In contrast, family members ($\beta = 0.108$, $p = 0.073$) did not significantly impact the intention to continue product collection purchases.

Novel to this study, friends, social media, and romantic partners were found to influence the intention to purchase products for their collections. Surprisingly, the family was not found to influence the intention to purchase products. To further investigate such influences, this study also included qualitative data analysis.

### 4.2. Findings

The open-ended survey questions were coded for themes using a grounded theory approach. The types of collections represented in this study are wide-reaching and include various products, including coins, fabric, toys, books, and virtual items. The 213 participants in this study had 391 collections. Coins were the most popular (45 collections), followed by books (32 collections), miniatures and figurines (30 collections), stamps (26 collections), wearables/fabrics (25 collections), pop culture memorabilia (24 collections), art (21 collections), and comic books (21 collections).

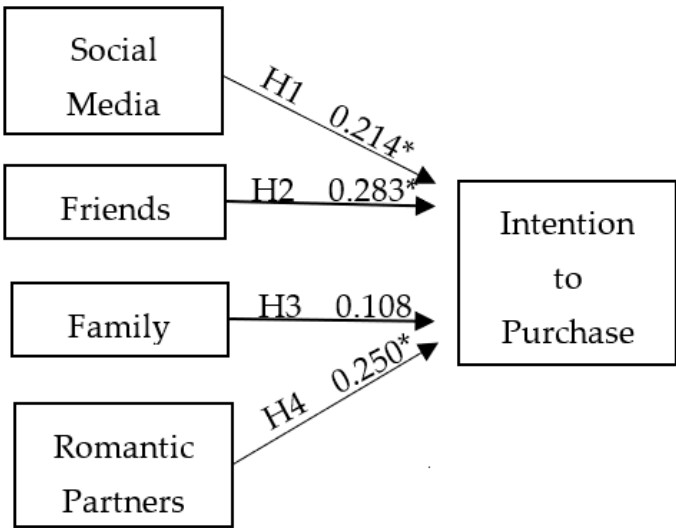

**Figure 1.** Results. Note. * *p* < 0.05.

4.2.1. Inspiration to Collect

Participants were asked, "Who (or what) inspired you to start collecting?", which yielded extensive qualitative data. After coding each participant's response, seven different themes emerged. However, three themes were overwhelmingly common among participants, which included gaining inspiration from personal interests, family members, and friends.

One hundred and two mentions were made on how participants were personally driven to build a product collection. Some participants were drawn to a specific topic or product category based on lived experiences, as demonstrated through the following quote:

> "It was because I caught living frogs as a kid, and then I would release them right after catching them. I just loved them."

Other participants were somewhat surprised to have a product collection, as the collection grew slowly and was not intentional.

> "No one, in particular, inspired me, nor any particular event. It just sort of happened. I found that it made me happy to collect, gave me a sense of nostalgia and also a sense of happiness and calm as I looked around and saw the things I enjoyed, that made me happy, surrounding me in my home."

Interestingly, a participant highlighted that the collection might be a wonderful gift to pass down to family members, indirectly echoed by the inheritance of other people's collections.

> "I decided to collect them so I could pass them down to my children."

Other participants mentioned that they received ample gifts from others, which spurred the collection's growth.

> "I sort of fell into it. I had a few that I liked, but then my family started giving them to me for birthdays and holidays."

Family members were also cited as one of the biggest inspirations to start a product collection, as 102 participants mentioned how family played a role. Many participants discussed how the collection brought their loved ones together.

> "My brother had a stamp collection, and he would spend time with me, explaining where each stamp was from. Stamp collecting brought me closer to my brother, whom I really enjoyed, spending time with."

Similarly, participants mentioned that family members had started collections, which the receiver continued over time. Some collections were explicitly started for the receiver; others were inherited.

"My parents started collecting key chains for me when I was a baby, and I decided to keep it going because I love how unique and prevalent key chains are. You can find cute key chains everywhere. They are small enough where they don't take up too much space. All of my 200 key chains fit nicely in two medium-size storage boxes, so it's not like my collection is ruining my life."

Many participants mentioned inheriting a collection from a family member that has passed.

"My aunt had a ton of baseball programs. I now have most of her collection since she passed a few years ago."

Interestingly, one participant said that a family member was the only reason the collection was started.

"I was not really that inspired. It was my mother that was kind of excited about this collection."

Fifty-two participants mentioned friends as an inspiring force for building a product collection. All participants who said "gaining inspiration from friends" cited that friends were the ones who shared information about the product or that the product helped the participant fit into a specific social group.

"My friends all had Pokemon, once I got my first deck, I just couldn't stop buying them."

Participants mentioned a few categories beyond personal, family, and friend inspirations. However, celebrities (10 mentions), social media (7 mentions), brands/cartoons (6 mentions), and romantic partners (4 mentions) were discussed by participants.

The quote below highlights how childhood cartoons drove the collection to share how brands/cartoons served as inspiration.

"I often watched Hello Kitty reruns on Toon Disney in the summer when I was a young girl. Hello Kitty became my favorite character."

Information about how romantic relationships inspire collections was minimal. However, the inspiration typically took place when feelings of nostalgia emerged.

"My spouse used to dress up as Santa for the children's holiday program at our local public, and that sparked an interest in collecting vintage Santas."

Various social media platforms were mentioned when discussing the inspiration behind collections. Facebook and YouTube were primarily addressed, as people would see the products on social media and would be interested in starting such collections.

Celebrities were also found to inspire collections, building on fandom consumption-like behavior.

"Matt Kenseth was a local race car driver that was talented enough to break into NASCAR and even became a champion. That got me collecting."

### 4.2.2. Starting the Collection

Participants were asked, "How did you get started in collecting?" After coding each participant's response, 101 mentions were made, indicating the collection happened gradually. In comparison, 50 mentions stated that their family members helped them get started, and 25 stated that the collection began with a gift. In contrast, only 22 mentioned having peer support in creating a collection, 8 mentioned starting collections during a life change, and only 2 mentioned a romantic partner's support. Interestingly, 14 mentions were made of using an online forum for information to start a collection.



### 4.2.3. Duration of Collection

Due to the open-ended nature of the question "When did you start collecting?" responses were varied in nature. Responses included either the specific number of years one has been collecting or whether the collection started in childhood or adulthood. Overall, most collections began in childhood, as 105 participants mentioned an early beginning to a collection, while 27 participants mentioned that the collection started in adulthood. Many of the responses that involved a specific number of years of collecting indicated that most collections were created in childhood or early adulthood.

### 4.2.4. Motivations to Collect

When asked, "Why do you collect the type of product(s) you collect?" a variety of data was obtained. After coding each participant's response, six different themes emerged. Over 111 mentions were made about loving to have the product collection, indicating that the collection fulfills their hedonic needs. Second, 47 mentions were made of how the product collection brings back memories and nostalgia. A total of 28 mentions were made of how the collection is either entertaining or provides knowledge, while 16 mentions were made that family and friends were the reason why the person was collecting. Interestingly, 14 found their collections financially valuable, and 7 mentions were explicitly made about the collection's utility in their lives.

### 4.2.5. Locating Products for Collection

Participant responses were coded based on their responses to the question, "Where do you typically buy products for your collection?" which yielded a wide range of answers. There were 119 mentions of obtaining products at specialty stores, including retailers focused on specific product categories, such as hobby stores and gift shops. Distinct entities were also mentioned, including post offices (seven mentions, typically for stamps) and banks (one mention, typically for coins). Many people also found products for their collection in antique shops (23 mentions), secondhand stores (35 mentions), or garage sales (13 mentions). Tradeshows (12 mentions) and art galleries (4) also had followers. Twenty-two people mentioned that they only received products for their collection via gifts from others.

Digitally, many participants also found products online (184 mentions), while 44 specifically mentioned eBay, 36 mentioned Amazon, and 10 mentioned Etsy. Seven people spoke about finding products on Facebook, while one specifically mentioned using a mobile app.

### 4.2.6. Impact of Technology on the Collection

Participants were also asked, "Has technology altered the way that you collect?" in which 182 participants said that technology altered how they collect. A total of 98 participants said technology played no role, while 12 participants stated that technology somewhat changed how they collect.

Since online retailing has emerged, some have mentioned how products can be easily found online.

> "Well, when I was a kid, buying packs at a physical store was pretty much the only way to get cards. So the growth of online stores has definitely changed the way you can collect them. Like I mentioned earlier, the website tcgplayer.com enabled me to buy the specific cards I'm missing, rather than gambling on packs."

Participants can easily find rare or other wanted products.

> "Technology has helped me to collect cards/items that are no longer available to sell in stores due to they stopped selling them years ago. It's better online so I don't have to go to garage or yard sales to find these items."

Some are tied to what the product is.

"Technology has not really altered the way I collect. I did buy a vintage 1990s pack of key chains from a lady in my town on offerup.com. That was the first time I bought key chains from someone directly. I usually buy them at gift shops when I travel. I could easily go nuts on eBay and buy $$$ worth of key chains, but I prefer to buy them when I am traveling."

There are some challenges when collecting digital items.

"Yes, it has expanded my collection to the digital realm, which I, unfortunately, can't organize like I can with my physical collection."

Others used technology to help organize the collected products, including Microsoft Word (Version 2310, Microsoft 365, Redmond, WA, USA) and other organization software.

A few participants also highlighted that they have used technology their entire lives, so technology has always played a role.

"I often see news about a discontinued product, but I've had internet all my life, so. . ."

As a true sign of the times, a participant highlighted how technology allowed the collection to continue while bricks-and-mortar stores were closed during the COVID-19 pandemic.

Participants also found additions to their collections online, including on eBay and Amazon. For some specific types of collections, specialty stores (e.g., gift shops, toy stores) and second-hand stores were frequented.

### 4.2.7. Summary of Themes

Based on the themes emerging from the data, an overarching theme of nostalgia can be found. Participants connected with products based on memories and collections were primarily built on personal interest in products or receiving gifts from family. Collections focused on personal interests were based on product affinity, lower price points, and memory-invoking products. Collections started as gifts received from family included the acquisition of family heirlooms or products focused on the person's interests. Overall, this overarching theme is also linked to family members who typically have a role in initiating a collection. Still, that influence lessens over time, while other influences, including friends, romantic partners, and social media, influence the collector after the collection has started.

## 5. Discussion

Individual motivations have been found to play a role in building product collections. In the qualitative portion of this study, 47.88% of participants mentioned how they personally wanted to build their collection from the start, and 52.11% shared that they loved having the collection. A major theme indicated that product collections might instill a feeling of nostalgia and memories. Many participants also suggested that the product collection grew over time and was not originally intended to be a collection. As collectors continue their collection, it has been found that the focus becomes on constructing the whole instead of individual pieces (Mueller 2019).

Most products for collections are purchased at specialty stores. Collectors may be subject to the halo effect, where the presentation of the product may elicit more sales (Mueller 2019). However, 85.45% of participants also used technology to learn about products to add to their collections. If an arousal state of the collector is reached when investigating the product, the purchase decision may be made. Overall, humans tend to value things we own more than things we do not (Mueller 2019).

### 5.1. Family

This study identified family members' impact, highlighting how family plays a role at different times during the collection process. The quantitative results indicate that family members do not significantly impact the intention to continue product collection purchases. However, the qualitative findings show that family members are the biggest inspiration

when starting the product collection, as indicated by 47.88% of participants. The influence of family members has been well documented, as children are socialized to be consumers by observing family members (Venn et al. 2017). Families have also been previously found to be influential in starting the collection process, as families tend to purchase products to fulfill children's interests (Gao et al. 2014). However, parallel to this study, family members have been found to influence purchase attitudes but have not been found to influence purchase intention (Johnson Jorgensen and Ha 2019).

### 5.2. Friends

When making purchases for existing product collections, friends were the most impactful. Similarly, friendships relate to a greater purchase intention than strangers (Li et al. 2018; Zhu et al. 2016). Social circles have also been linked to the rate of product adoption and peer-related purchases. In contrast, if someone has a lot of friends in their network, it is less likely for the purchase to be made (Zhu et al. 2016). However, if friends own a piece of a collection, past research may not be applicable, as collected products may be worth less if others possess a piece of a collection (Mueller 2019).

### 5.3. Romantic Partners

While friends are the most impactful on the purchase intention of continuing product selections, romantic partners were the second most significant influence. Previous research also supports the impact of romantic partners on purchasing, as Yu (2011) determined that relationship strength, satisfaction, and length of the relationship predict shopping behavior. People engaged in a romantic crush have been found to seek product variety (Huang and Dong 2018), while people who are satisfied with their relationship avoid conspicuous consumption (Liu et al. 2020). Beyond the participants' self-reported impact of romantic relationships, limited experiences were shared beyond purchasing support.

### 5.4. Social Media

Social media plays a role in many aspects of people's lives. The results of this study indicate that social media impacts purchase decisions for collections. On a routine basis, people engage in social media to access information and socially connect (Shang et al. 2017). When collecting, gaining knowledge about products in online forums and communities may provide emotional and informational support (Riaz et al. 2021). People using social media intensely tend to have greater perceived usefulness and purchase intention (Harrigan et al. 2021). It has also been found that social media posts and electronic word-of-mouth influence attitudes and purchase intentions (Shang et al. 2017; Johnson Jorgensen and Ha 2019).

### 5.5. Study Summary

Consistent with the current study, Lafferty et al. (2013) and Nordsletten and Mataix-Cols (2012) found that collecting typically starts early in life and is influenced by parents and peers. However, this study also highlights the role family plays during different collection stages and the investigation of possible social media and romantic relationship influences. Thus, it has been determined that friends, romantic partners, and social media influence the intention to purchase products for the collection. In contrast, family members have been found to play a role at the beginning of the collection. Qualitative insights indicate that nostalgia or documentation of memories is one of the biggest motivations behind collections.

### 5.6. Future Directions, Limitations, and Implications

To date, little research has investigated the influences on product collections. This study aimed to fill that gap by determining how traditional influences (friends, family, romantic relationships) and technology (social media) play a role in current collection processes. To illustrate these connections between the quantitative and qualitative phases

of this study, a joint analysis is available in Appendix B. Novel to the literature in this field, it has been determined that family influence tends to happen early in the product collection process. At the same time, friends, romantic relationships, and social media play a role after the collection has started. Based on the results and findings of this study, a model for future inquiry was developed and is available in Figure 2. This model highlights the family's distinct role in initiating a collection, while other influences occur later. While this study serves as a foundation for research on this topic, studies on the motivations of collectors should be completed to gain additional insights from more diverse populations. Additional research is also needed to investigate how the transition from family influences to friend, romantic, or social media influences unfolds. In addition, future research should include exploring and explaining virtual collections compared to physical collections.

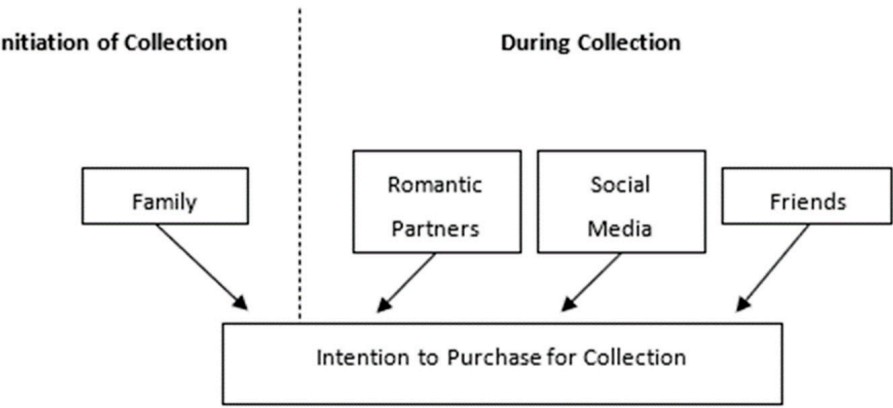

**Figure 2.** Model for future inquiry.

Limitations also exist for this study. A convenience sample of participants was employed, which yielded a participant sample that skewed more toward females, Caucasians, and people between the ages of 25–44. A random sample may have produced a more representative sample. Also, recruitment occurred on Amazon's MTurk, which only recruits a subset of the population. Despite Amazon MTurk's limitations, Kan and Drummy (2018) outlined some concerns that may arise on MTurk, including deception by participants, lack of environmental control, and routine exposure to research procedures. Due to the small compensation and online control of taking this study's survey, participants are unlikely to engage in deceptive practices. However, future studies would benefit from recruiting from other platforms.

The implications for this study are primarily in two distinct areas. First, marketers, retailers, and collection-related businesses should consider family members' role at the start of a collection and the influences that friends, romantic partners, and social media have when someone continues their collection. Creating targeted marketing campaigns based on the stage at which their products fall along the collection timeline may result in increased profits. Second, collectors and museums can identify at which stages collections may be built or transferred to another owner at the end of the collection cycle. Ultimately, collections serve as a part of people's lives and stories.

## 6. Conclusions

Emerging from this research, it was found that family members play a unique role when individuals collect products. Interestingly, family members were instrumental when individuals started their collections but were not found to play much of a role after the collection was created beyond gift-giving. Once the collection was underway, it was found that friends, romantic partners, and social media were influential in continuing product collection purchases. Based on this finding, additional research is needed to understand how the influence of family members on product collections subsides over time and how the other influences emerge to support the individuals' collections.

Collecting products serves as a niche market in today's retail industry. Thus, this research provides insight for retailers, marketers, and collection-based businesses to observe the motivations behind their consumers, which can lead to increased reach and profits. This study also adds to the knowledge of collector socialization, which has been very limited (Gao et al. 2014) due to foci on specific collection categories (e.g., coins, etc.) or the difference between collectors and hoarders (Nordsletten and Mataix-Cols 2012). Therefore, this study bridges the previous gap in the knowledge of collector socialization and further supports a need for future research on this topic.

**Author Contributions:** Conceptualization, J.J.J. and K.S., methodology, J.J.J. and K.S.; formal analysis, J.J.J. and K.S.; resources, K.S. and M.S.; data curation, J.J.J. and M.S.; writing—original draft preparation, K.S. and J.J.J.; writing—review and editing, J.J.J. and M.S.; supervision, J.J.J. All authors have read and agreed to the published version of the manuscript.

**Funding:** This research received no external funding.

**Institutional Review Board Statement:** The study was conducted in accordance with the Declaration of Helsinki, and approved by the Institutional Review Board (or Ethics Committee) of the University of Nebraska-Lincoln (protocol code 20220221783EX and date of approval of 18 February 2022).

**Informed Consent Statement:** Informed consent was obtained from all subjects involved in the study.

**Data Availability Statement:** The data presented in this study are available on request from the corresponding author. The data are not publicly available due to the individualized qualitative nature of the data.

**Conflicts of Interest:** The authors declare no conflict of interest.

## Appendix A

**Table A1.** Survey Questions.

| Variable | Survey Question |
| --- | --- |
| Quantitative Questions | |
| Online social networks | I spend a lot of time using online social networks (i.e., Pinterest, Facebook, Instagram) to look at products before purchasing an item for my collection. [1] <br> I spend a lot of time talking with my online social network friends about purchasing an item for my collection. [1] <br> Pictures from online social networks encourage me to make purchases for my collection. [1] <br> My online social network friends encourage me to make purchases for my collection. [1] <br> Online social networks help me find items for my collection. [1] <br> My online social network friends help me find items for my collection. [1] <br> I search online social networking websites for advice about buying products for my collection. [1] <br> I ask my online social network friends for advice about buying products for my collection. [1] |
| Family | My family encourages me to make purchases for my collection using my own judgment. [4] <br> My family and I tell each other where to find items for my collection. [2] <br> I ask my family for advice about buying products for my collection. [3] <br> My family encourages me to make purchases of items I want for my collection. [4] <br> I spend a lot of time talking with my family about purchasing an item for my collection. [5] |
| Friends | My friends encourage me to make purchases for my collection using my own judgment. [4] <br> My friends and I tell each other where to find items for my collection. [2] <br> I ask my friends for advice about buying products for my collection. [3] <br> My friends encourage me to make purchases for my collection. [4] <br> I spend a lot of time talking with my friends about purchasing an item for my collection. [5] |

**Table A1.** *Cont.*

| Variable | Survey Question |
|---|---|
| Romantic partners | My romantic partner(s) encourage me to make purchases for my collection using my own judgment. [4] |
| | My romantic partner(s) and I tell each other where to find items for my collection. [2] |
| | I ask my romantic partner(s) for advice about buying products for my collection. [3] |
| | My romantic partner(s) encourage me to make purchases for my collection. [4] |
| | I spend a lot of time talking with my romantic partner(s) about purchasing an item for my collection. [5] |
| Intention to purchase | I intend to buy products for my collection that I have searched for online in the future.[4] |
| | I intend to buy products for my collection that I have searched for on online social networking websites in the future. [4] |
| | I intend to buy products for my collection that family members recommend in the future. [4] |
| | I intend to buy products for my collection that my friends recommend in the future. [4] |
| | I intend to buy products for my collection that my romantic partner(s) recommend in the future. [4] |
| *Qualitative Questions* | |
| | What type of product(s) do you collect? Why? |
| | Where do you typically buy products for your collection? |
| | How did you get started in collecting? |
| | Who (or what) inspired you to start collecting? |
| | When did you start collecting? |
| | Has technology altered the way that you collect? |

[1] Adapted from (Johnson Jorgensen and Kean 2018); [2] adapted from (Mangleburg et al. 1997; Lueg et al. 2006; Johnson Jorgensen and Kean 2018); [3] adapted from (Moschis and Moore 1979; Lueg et al. 2006; Johnson Jorgensen and Kean 2018); [4] adapted from (Lueg et al. 2006; Johnson Jorgensen and Kean 2018); [5] adapted from (Bush et al. 1999; Lueg et al. 2006; Johnson Jorgensen and Kean 2018).

## Appendix B

**Table A2.** Joint Analysis.

| Socialization Agent | Quantitative Results | Qualitative Findings |
|---|---|---|
| Friends | Friends were the most impactful on the intention to purchase for a collection | Friends served as an information source for product collections<br>Impact of friends increases over time |
| Romantic partner | Romantic partners were found to significantly impact the purchase intention for a collection | A connection between romantic partners and nostalgia evoked from the product collection was found |
| Online social media | Online social media was found to significantly impact the intention to purchase for a collection, but less so than friends and romantic partners | Online social media was discussed the least in the qualitative phase of the study, but played a role in the acquiring of products for a collection<br>Online marketplaces for collected products (e.g., eBay, etc.) were found to be highly accessible |
| Family | Family was not found to significantly impact purchase decisions for a collection | Family was described as the inspiration behind starting a product collection<br>Family members were found to pass along product collections or collections were started based on gifts received<br>Influence of family lessens over time |

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
