# Peer review of "Motivations to Collect: How Consumers Are Socialized to Build Product Collections"

_socsci, doi:10.3390/socsci12120671_

Round 1
Reviewer 1 Report
Comments and Suggestions for Authors
I find the paper very interesting, as collecting products is a very rare topic among marketing publications. In my opinion the aim, methods, structure, conclusions are correct. The advantage of this study is to use mixed, both qualitative and quantitavive approach. I find the analysis performed in accordance with the rules of scientific work. There is good introduction, explaining the importance of the topic. Hypotheses are correctly derived from literature. Still, as it is practiced, they should be stadet in Present Simple not Future Simple tense, eg. Family impacts... rather then Family will impact... Then the model is developed, discussed and concluded in the end.
My only preservation is very limited information concerning the statistical model, which is extremally limited here. First of all, The authors should include the questionnaire as an appendix (it is only mentioned that there were 34 questions). Then, the indices describing model fit should be presented and discussed (only the range of Cronbach alfa for the scales, R2, F are given, and p-values for each Beta), and this is not enough. The authors should analyse and describe how precisely the proposed model reflects the researched domain.
Reviewer 2 Report
Comments and Suggestions for Authors
The paper is an excellent contribution to the field of consumer behavior science and marketing research. The authors have done a good job in conducting thorough research and presenting their findings in a clear and concise manner. One of the strengths of this paper is the comprehensive literature review that provides a solid foundation for the research. It not only highlights the significance of the topic but also underscores the current gaps in knowledge, setting the stage for the study's objectives. Cited sources are all related to the topic Although, I would recommend adding some newer sources, published in 2023. The methodology used in this research is robust and well-explained, ensuring the reproducibility of the study, which is crucial in scientific research. The data analysis and results section of the paper are well elaborated. The authors have effectively used figures, and statistical analysis to convey their findings. The presentation of the data is not only informative but also visually appealing, making it accessible to a wide range of readers, from experts in the field to those with a general interest in the topic. Furthermore, the discussion and conclusion sections of the paper provide valuable insights into the implications of the study's findings. The authors have successfully linked their research to real-world applications, emphasizing the importance of studying consumer behavior. This aspect of the paper is particularly commendable, as it highlights the future directions of the research. In terms of writing style and organization, the paper is well-written. The language is clear and concise, and the flow of the paper is logical, making it easy to follow the argument from introduction to conclusion. The references are also extensive and well-cited, demonstrating the thoroughness of the literature review. Overall, this paper is a commendable piece of research that significantly advances our understanding of researched topic.
Reviewer 3 Report
Comments and Suggestions for Authors
Abstract
The objective of this research article has been well expressed in the abstract itself, which is to analyze the motivations behind people’s collection of various products. The extent to which consumers are socialized in order to build product collections is a matter that has been touched upon in the abstract, and which also serves as the key theme of the article. The type of methodology that has been used to carry out this work, namely, the use of qualitative research methodology and an overview of the research results have also been summed up in the abstract of the article.
Introduction
The introductory section of this research article begins by providing an overview of what product collections entail, and it makes a reference to the fact that people have a penchant for collecting digital and physical products over a period of time. The nature and type of products that people collect in their everyday lives, have been made known in the introduction, and the need for studying this topic from a sociological perspective has also been highlighted in the introduction to the piece.
Literature Review
The literature review that was conducted for this article makes a reference to the various theories and arguments that have been put forward by scholars regarding the habits and the behavioral traits of collectors, and the nature of the items that they collect, either out of fancy or because they are interested in conducting a research project with the same. It has been made known in the literature review that scholarship on the subject of collector socialization is quite limited and which is why there is extensive research that needs to be conducted in this area of study. The motivations and the interests of hoarders is a topic that has been studied at great length by scholars and this is something that has been analyzed and deliberated upon in the literature review. The gaps in the review of literature have also been identified and the purpose or the rationale behind conducting this particular study has been put forward towards the end of the literature review.
Methodology
There is a mixed methods approach that was adopted for undertaking this study. Both qualitative and quantitative data were obtained through the use of surveys, that entails answering questions using a Likert Scale. The quantitative data obtained from the survey responses was analyzed using SPSS while the qualitative data was analyzed using the well-known grounded theory approach.
Results
There are 2 specific reasons why people collect items or feel motivated to collect products, as has been made known through the findings of this study. Firstly, they are encouraged by their family members to do so and feel inclined to collect items because they have seen members of their family do the same and have even inherited such items from them. Secondly, collector’s items have a niche market, and there is a lot of profit that one can incur through the sale of collector’s items.
Conclusion
In conclusion, consumers are motivated to collect items because of family traditions and also because of the market value of collector’s items.
Comments
This is a study which has been conducted on the strength of primary and secondary research of a quantitative and qualitative nature. However, there is more extensive research that ought to be carried out on the subject, using the same methodology to gather more inputs about the motivations behind people to collect items.
Comments on the Quality of English LanguageMinor editing of English language required
Reviewer 4 Report
Comments and Suggestions for Authors
The subject under study is to know the social reasons that motivate collectors.
It is an interesting topic and the proposal is attractive. The document is well written and the references are generally attractive and appropriate for a scientific article.
The major problem of the study is the methodology. It is too bare-bones and there are missing elements to add to it. It should be carried out with the three classic sections of a research methodology: research design, data collection, and data analysis. In each of them, mention should be made of the quantitative study on the one hand and the qualitative study on the other.
In this methodology, information should be added: in the quantitative part, the date on which the study was carried out, more information about how the sample was obtained and other studies that have used a similar analysis with these questions should be added. On the qualitative side, more information is needed about the date on which the study was conducted, explaining more about the Grounded Theory approach and explaining more about the content analysis carried out.
On the results, some information is also missing, the results of the quantitative analysis are scarce, and more depth should be given to the results obtained. Regarding the qualitative research, a table specifying the categories obtained would be useful.
